# Abnormal Splicing of *GALC* Transcripts Underlies Unusual Cases of Krabbe Disease

**DOI:** 10.3390/biomedicines13123114

**Published:** 2025-12-17

**Authors:** María Domínguez-Ruiz, Juan Luis Chico, Laura López-Marín, Sinziana Stanescu, Pablo Crujeiras, Daniel Rodrigues, María-Elena de las Heras-Alonso, Rosana Torremocha, María del Mar Meijón-Ortigueira, Patricia Muñoz-Díez, Val F. Lanza, Cristóbal Colón, Jesús Villarrubia, Amaya Belanger, Francisco J. del Castillo

**Affiliations:** 1Servicio de Genética, Hospital Universitario Ramón y Cajal, IRYCIS, 28034 Madrid, Spain; mdominguezr@salud.madrid.org; 2Centro de Investigación Biomédica en Red de Enfermedades Raras (CIBERER), 28034 Madrid, Spain; 3Servicio de Neurología, Hospital Universitario Ramón y Cajal, IRYCIS, Universidad de Alcalá de Henares, 28034 Madrid, Spain; juanluis.chico@salud.madrid.org; 4Sección de Neuropediatría, Hospital Infantil Universitario Niño Jesús, 28009 Madrid, Spain; llopezm@salud.madrid.org; 5Servicio de Pediatría, Hospital Universitario Ramón y Cajal, IRYCIS, 28034 Madrid, Spain; sinziana.stanescu@salud.madrid.org (S.S.); amaya.belanger@salud.madrid.org (A.B.); 6CSUR de Enfermedades Metabólicas, European Reference Network for Hereditary Metabolic Disorders (MetabERN), Hospital Universitario Ramón y Cajal, IRYCIS, 28034 Madrid, Spain; jesus.villarrubia@salud.madrid.org; 7Congenital Metabolic Diseases Unit, Department of Neonatology, University Clinical Hospital of Santiago de Compostela, Instituto de Investigación Sanitaria de Santiago (IDIS), European Reference Network for Hereditary Metabolic Disorders (MetabERN), 15706 Santiago de Compostela, Spain; pablo.crujeiras.barral@sergas.es (P.C.); daniel.caiola.candeias@sergas.es (D.R.); cristobal.colon.mejeras@sergas.es (C.C.); 8Centro de Investigación Biomédica en Red Enfermedades Raras (CIBERER), 15706 Santiago de Compostela, Spain; 9Servicio de Dermatología, Hospital Universitario Ramón y Cajal, IRYCIS, 28034 Madrid, Spain; elena.heras@salud.madrid.org; 10Unidad de Genómica, Fundación Parque Científico de Madrid, 28049 Madrid, Spain; rosana.torremocha@fpcm.es; 11Servicio de Hematología, Hospital Universitario Ramón y Cajal, IRYCIS, 28034 Madrid, Spain; mmar.meijon@salud.madrid.org (M.d.M.M.-O.);; 12UCA de Genómica Traslacional y Bioinformática (UCA-GTB), Hospital Universitario Ramón y Cajal, IRYCIS, 28034 Madrid, Spain; val.fernandez@salud.madrid.org; 13Centro de Investigación Biomédica en Red de Enfermedades Infecciosas (CIBERINFEC), 28034 Madrid, Spain

**Keywords:** rare diseases, lysosomal diseases, NGS, Krabbe disease, globoid cell leukodystrophy, functional assays, phenotypic variation, deep intronic variants

## Abstract

**Background/Objectives**: Krabbe disease (KD) is a hereditary lysosomal disorder whose hallmark is progressive demyelination, with variable involvement of the central nervous system. It is caused by pathogenic variants in the *GALC* gene that disrupt the function of its gene product, the lysosomal enzyme galactosylceramidase. We analyzed two unrelated cases (one early infantile and one adult) with a clinical suspicion of KD. **Methods**: We used a combination of biochemical techniques (high-performance liquid chromatography–tandem mass spectrometry), NGS (resequencing gene panels), splicing assays, and molecular modeling to identify and analyze the pathogenicity of the variants underlying the disorder. **Results**: The two probands were compound heterozygotes for disease-causing variants in the *GALC* gene, encoding the lysosomal hydrolase galactosylceramidase. Three of the variants were novel and caused aberrant splicing, either by exon skipping (c.908+5G>A and c.1034-1G>C) or by inclusion of a cryptic, deep intronic pseudoexon (c.621+772G>C). The fourth variant was a known missense change (c.956A>G, p.(Tyr319Cys)) with conflicting interpretations of pathogenicity in the databases. **Conclusions**: We demonstrated the pathogenicity of the three novel splicing variants, all with strong impact on galactosylceramidase function. We also concluded that the c.956A>G missense variant is a hypomorph usually underlying the later-onset, milder phenotypes of KD. Our results stress the importance of integrated approaches combining clinical, biochemical, and genetic testing to obtain a definitive diagnosis of lysosomal diseases.

## 1. Introduction

Lysosomal diseases (LDs) are a group of about 70 rare hereditary disorders (combined incidence: about 1 in 5000–5500 newborns) caused by lysosomal dysfunction due to pathogenic variants in the genes encoding lysosomal hydrolases, transporters, receptors, or enzyme activators [1]. These disorders are usually syndromic in nature, with symptoms and signs in different bodily systems, including the central nervous system (CNS). For those LDs that are due to enzyme deficiencies, diagnosis relies, upon clinical suspicion, on assaying the activity of the enzyme in patients’ cultured fibroblasts or lymphocytes [2] or, much more frequently, in dried blood spots (DBSs) [3], followed by genetic analyses to identify the underlying pathogenic variants [1].

Krabbe disease (KD), one of these LDs, is due to a deficiency of the lysosomal hydrolase galactosylceramidase (EC 3.2.1.46, also termed beta-galactocerebrosidase), which removes the galactose moiety of galactosylceramides by breaking the galactose ester bonds. KD is also termed globoid cell leukodystrophy because of the eponymous large, abnormal multinucleated macrophages that store undegraded galactosylceramide lipids and that are observed in the brains of KD patients [4]. In the CNS, galactosylceramidase activity is essential for recycling its target lipids during myelin turnover. The deficiency of galactosylceramidase hampers remyelination, though myelin turnover continues [5]. Thus, a hallmark of KD is progressive demyelination, leading to neurological deficits of variable severity. As with most LDs, the disease may appear at any age, from the early infantile period to adulthood (Appendix A), though phenotype and disease severity are correlated with age of onset [6].

Early infantile presentation is the most common form of KD (up to 85–90% of cases), with relatively homogeneous symptoms and course [6]. It starts with developmental delays (noticed before the age of six months), diverse symptoms and signs attributable to CNS dysfunction (such as irritability, poor head control, stiffness, etc.), and peripheral neuropathy. Rapid clinical deterioration with increasingly severe neurological signs (which may include optic atrophy, deafness, motor disabilities, poor feeding requiring tube feeds, etc.) eventually leads to death in nearly all patients, most usually before two years of age, because of end-stage morbidities (respiratory failure, infections, emaciation) [4,6]. However, later-onset forms (late infantile, juvenile, adult) have wider variability in presenting symptoms and natural history and are thus much more difficult to recognize. Indeed, the very rare adult form of KD is usually misdiagnosed initially because it may masquerade as spastic paraplegia [7,8] or multiple sclerosis [9,10].

KD is caused by biallelic pathogenic mutations either in the *GALC* gene (OMIM 606890, on chromosome 14q31.3) [11], encoding galactosylceramidase, or in the *PSAP* gene (OMIM 176801, on chromosome 10q22.1), which codes for a precursor protein cleaved, upon reaching the lysosome, into four saposins, A to D. Inactivation of galactosylceramidase causes KD proper (OMIM 245200), whereas inactivation of saposin A, the essential co-activator that extracts galactosylceramide lipids from the membrane to present them to the hydrolase [12], results in KD due to saposin A deficiency (OMIM 611722). The mutation spectrum includes over 260 mutations in *GALC* and 3 mutations in *PSAP* [13]. Most of them are private, though some variants, such as the common 30-kb deletion named c.1161+6532_polyA+9kbdel [14], have been reported in homozygosity or compound heterozygosity in multiple patients. Broadly speaking, the impact of each mutation on galactosylceramidase activity correlates with disease severity and age of onset, with null variants (such as c.1161+6532_polyA+9kbdel) usually involved in the very severe, early infantile presentation [15]. However, intrafamilial variation has been reported among individuals with the same genotype and markedly different severities and ages at onset [6]. This suggests that other genetic and environmental factors modulate the clinical presentation of KD.

Here, we report on two unrelated cases of KD with early infantile and adult onset, respectively. We describe the underlying disease-causing variants, demonstrate their pathogenicity, and discuss their effects on clinical presentation.

## 2. Materials and Methods

### 2.1. Subjects

Our project was approved by the Ethical Committee of Hospital Universitario Ramón y Cajal with approval code 249-23, in accordance with the 1964 Declaration of Helsinki. We obtained written informed consent from all participating subjects or their parents.

Patients received clinical examination as part of their routine clinical care. We obtained clinical data from their medical histories and through reappraisal after a genetic diagnosis was established.

### 2.2. Biochemical Analyses

Biochemical analyses were carried out at the Congenital Metabolic Diseases Unit of the Santiago de Compostela University Clinical Hospital. Diagnosis of KD consists of a β-galactosylceramidase enzymatic assay, carried out on DBS samples by UPLC-MS/MS (ultraperformance liquid chromatography–tandem mass spectrometry). We analyzed a 3.2 mm DBS punch with the NeoLSD™ MSMS Kit (Revvity 3093-0020, Waltham, MA, USA) following a 20 h incubation period. Enzymatic activity was quantified by UPLC-MS/MS using a Sciex 4500 MD QTRAP instrument (SCIEX, Framingham, MA, USA). Multiple reaction monitoring (MRM) in positive electrospray ionization (ESI) mode was employed to detect both the *GALC* gene product (*m*/*z* 412.1 → 264.2) and the internal standard (*m*/*z* 417.1 → 264.2). Detection was performed via flow-injection analysis (FIA) using an isocratic flow at 0.12 mL/min over 1.0 min.

The analysis of lysosphingolipids was performed using a modified protocol based on Rolfs et al. [16]. Briefly, 50 µL of plasma or standard solution containing lyso-globotriaosylsphingosine (Lyso-Gb3, Cayman Chemical Company 24873, Ann Arbor, MI, USA), glucosylsphingosine (Lyso-Gb1, Matreya LLC, 2086), and lyso-sphingomyelin (Lyso-SM, Cayman Chemical Company 10007947) [17], at concentrations ranging from 0 to 1250 ng/mL, were mixed with 100 µL of internal standard (lactosylsphingosine (Lyso-Gb2, Matreya LLC 1517, State College, PA, USA), 10 ng/mL) in a 96-deep-well plate (Thermo Fisher Scientific, Waltham, MA, USA). Following agitation at 1000 rpm for 5 min at room temperature, the plate was centrifuged at 2200× *g* for 5 min, and the supernatant was transferred to a 96-well plate (Greiner, Kremsmünster, Austria). A 10 µL aliquot of each sample was analyzed by high-performance liquid chromatography–tandem mass spectrometry (HPLC-MS/MS) using a Sciex 4000 QTRAP instrument (SCIEX, Framingham, MA, USA). Detection of lysosphingolipids (Lyso-Gb3, *m*/*z* 786.6 → 282.4; Lyso-Gb1, *m*/*z* 462.4 → 282.0; Lyso-SM, *m*/*z* 465.4 → 184.2) and the internal standard (Lyso-Gb2, *m*/*z* 624.5 → 282.4) was performed using MRM in positive ESI mode. Separation by chromatography was performed on a Poroshell 120 EC-C18 column (2.7 µm, 3.0 × 50 mm; Agilent Technologies, Santa Clara, CA, USA) maintained at 45 °C. The mobile phases consisted of solvent A (50 mM formic acid in water) and solvent B (50 mM formic acid in a 1:1 [*v*/*v*] mixture of acetonitrile and acetone). The gradient elution, at a flow rate of 0.45 mL/min, was as follows: 0.00–1.00 min, 30% B; 1.00–3.00 min, linear increase to 100% B; 3.00–4.50 min, 100% B; and 4.50–5.50 min, re-equilibration at 30% B. All chemicals were LC-MS grade.

### 2.3. Genetic Techniques

#### 2.3.1. Nucleic Acid Extractions

Genomic DNA was extracted from peripheral blood samples collected in EDTA tubes with FlexiGene DNA (Qiagen, Hilden, Germany) at the Core Research Genomic Facility (UCA-GTB) of our hospital.

Total RNA was isolated from peripheral blood collected in PAXGene tubes (BD, Becton Dickinson, East Rutherford, NJ, USA) and purified with PAXgene Blood RNA products (PreAnalytiX, Hombrechtikon, Switzerland; Qiagen, Venlo, the Netherlands; and BD, Franklin Lakes, NJ, USA). Reverse transcription to obtain cDNA was performed using 500 ng of total RNA, random hexamer primers (pdN_6_), and SuperScript II reverse transcriptase (Invitrogen, Waltham, MA, USA) according to the manufacturer’s instructions.

#### 2.3.2. Next-Generation Sequencing (NGS) Techniques

Next-generation DNA sequencing was performed with the STP v3 gene panel [18,19], designed in our laboratory at the Servicio de Genética of our hospital. This gene panel targets 101 genes (Appendix A) that are known to be implicated in lysosomal diseases and their differential diagnoses, covering 498,932 bp of genomic sequence. The panel uses the SureSelect XT probe capture system from Agilent Technologies (Santa Clara, CA, USA). Specifically, the STP v3 gene panel includes *GALC* (OMIM 606890) and *PSAP* (OMIM 176801), both of which are involved in different forms of KD. The scope of STP v3 for each of the targeted genes includes all protein-coding exons, all exons for 5′-untranslated region, all the exon–intron boundaries, plus any other sequences, such as promoters or deep intronic regions, in which any pathogenic variants were as of 30 October 2018 at the ClinVar database [20].

We sequenced capture-enriched libraries on an Illumina MiSeq machine (Illumina, Inc., San Diego, CA, USA) at Fundación Parque Científico de Madrid. We mapped sequence data with novoAlign (Novocraft, Petaling Jaya, Selangor, Malaysia) against the GRCh37/hg19 human reference genome, and variant detection was performed with VarScan2 [21]. The panel was validated by sequencing a trio consisting of father (NA12891), mother (NA12892), and daughter (NA12878) from the reference 1463 family from the CEPH, with very high sensitivity (99.67%), specificity (99.99%), and PPV (98.22%). The panel complies with the laboratory standards established by the American College of Medical Genetics for NGS [22] because we obtained a median depth across all regions of 411X. We used the same conditions as the validation runs for the experiments reported here.

Qiagen Clinical Insight Interpret Translational (Qiagen, Hilden, Germany) was the tool used to annotate, filter, classify, and prioritize single-nucleotide variants and copy number variants. We successively filtered for allelic fraction (AF) (variants with AF < 20% were excluded), population frequency (PF) in the gnomAD [23] and 1000 Genome [24] databases (excluding any variants with PF ≥ 1%), functional consequences (retaining only variants that are missense, nonsense, loss of start or of stop codons, frameshifts or in-phase insertions or deletions, copy-number variants, targeting the 5 intronic nucleotides contiguous to the exon–intron boundary, or with PHRED-scaled CADD values ≥ 15) and classification by ACMG criteria [25] (only retaining pathogenic, likely pathogenic, or variants of unknown significance), as applied by Qiagen Clinical Insight Interpret Translational.

#### 2.3.3. Validation of Variants in *GALC* Detected in NGS Experiments

Any variants that passed all aforementioned filters within a gene involved in a known neurologic phenotype were selected for validation. We confirmed such variants and studied their segregation in the pedigrees with Sanger DNA sequencing, excluding any variants that did not segregate with the disease in the family. The primers and PCR conditions for amplicons appear in Appendix A. Sanger DNA sequencing was carried out in an ABI Prism 3100 Avant Genetic Analyzer (Applied Biosystems, Waltham, MA, USA).

#### 2.3.4. Detection of the Second Variant in *GALC* Monoallelic Cases

In individuals with just a single pathogenic variant identified in the *GALC* gene by our NGS analyses, we searched for a second pathogenic variant by multiplex-ligation probe amplification (MLPA) and by full-length cDNA analysis. MLPA was carried out with the commercial MLPA assay P446 (MRC-Holland, Amsterdam, the Netherlands), which includes probes for all exons of the *GALC* gene. For full-length cDNA analysis, we performed five overlapping PCRs on *GALC* cDNA (Appendix A). Two extra primer pairs (exons 7 to 9 and exons 5 to 10) were designed to evaluate the functional impact of the specific variant identified (Appendix A). Whenever agarose gel electrophoresis of cDNA PCR products revealed multiple bands, we excised and purified the bands using NucleoSpin Gel and PCR Clean-up (Macherey-Nagel, Düren, North Rhine-Westphalia, Germany), following the manufacturer’s instructions. Purified bands were then sequenced in an ABI Prism 3100 Avant Genetic Analyzer (Applied Biosystems, Waltham, MA, USA).

#### 2.3.5. Validation of Variants in *GBA1* Detected in NGS Experiments

Verification of identified *GBA1* gene variants was performed by long-range, gene-specific PCR, because *GBA1* has a highly similar, non-processed pseudogene, *GBA1LP*, which would otherwise be co-amplified and interfere with variant sequencing. We designed primers in sequence stretches that differed between the gene and the pseudogene (upper primer: 5′-CTT CCT AAA GTT GTC ACC CAT ACA TG-3′; lower primer: 5′-AAG CTC ACA CTG GCC CTG CT-3′), yielding an amplicon 6533-bp-long. Long-range PCR was performed using TaKaRa LA Taq DNA Polymerase (Takara Holdings, Kioto, Japan) with the following program: denaturation at 94 °C 2 min, 14 cycles of 94 °C for 30 s followed by an extension at 68 °C for 4 min 30 s, 16 cycles of 94 °C for 30 s and an extension at 68 °C for 4 min 45 s, and a final extension step at 72 °C for 10 min. We used a 1/20 dilution of the long-range amplicon as a template for a nested PCR targeting each *GBA1* exon of interest (Appendix A) for subsequent Sanger sequencing.

### 2.4. Assessment of the Pathogenicity of DNA Variants

To assess the pathogenicity of DNA variants, we used the guidelines from the American College of Medical Genetics and Genomics and the Association for Molecular Pathology (ACMG/AMP) [25], as implemented by both Qiagen Clinical Insight Interpret Translational and VarSome [26], using GRCh37/hg19 as the human reference genome. Functional predictions were obtained from dbNFSP v5.2 [27,28] (Genos Bioinformatics LLC, TX, USA). Detailed predictive analysis of the effect on the splicing of specific variants was performed with SpliceAI [29].

### 2.5. Assays for Effects on Splicing of GALC Variants c.621+772G>C and c.908+5G>A

For total RNA extraction, we obtained new blood samples from the parents of individual III:1 of family 42PET31, each heterozygous for one of the *GALC* variants under study. To analyze the effects on splicing of the paternal and maternal alleles, we used the primer pair targeting *GALC* cDNA “Exons 4 to 9” (Appendix A), with the forward primer labeled with the 6-FAM fluorochrome. PCR was performed on cDNAs from the mother, the father, and two additional wild-type controls. The PCR products were treated with T4 DNA polymerase (ThermoFisher, Waltham, MA, USA) to blunt any 3′ overhangs and ensure uniform fragment sizes. The AB SeqStudio Genetic Analyzer (Applied Biosystems, Waltham, MA, USA) was used for fragment sizing and quantification of the labeled PCR products. The amount of each amplicon was estimated from the area under the curve.

In parallel, we also performed PCR with unlabeled primers, and the products were analyzed by agarose gel electrophoresis. The corresponding bands were excised and sequenced by the Sanger method to verify amplicon identity.

### 2.6. Modeling the Effects of GALC Variant p.(Tyr319Cys) on Galactosylceramidase Structure

We retrieved from the RCSB Protein Data Bank [30] the following published X-ray diffraction structures: a galactosylceramidase monomer (3ZR5 and 3ZR6) [31], a galactosylceramidase monomer bound to substrate, covalent intermediate and product (4CCC, 4CCD, and 4CCE, respectively) [32], and the galactosylceramidase-saposin A heterotetrameric complex (5NXB) [12]. All of them correspond to murine galactosylceramidase. We also retrieved a high-confidence (>94%) model of the human galactosylceramidase monomer (AF-P54803-F1-v4) generated by AlphaFold2 [33] and included it in the AlphaFold Protein Structure Database [34,35,36]. We performed pairwise structural comparisons [37] between the AF-P54803-F1-v4 model and each of the published galactosylceramidase structures using the rigid-body protein alignment method TM-align [38] to verify the accuracy of the AlphaFold2 model.

Subsequently, we used this high-confidence model as a template to determine the effects of replacing the Tyr-319 residue with Cys. The altered amino acid side chain in the model was positioned with a backbone-dependent rotamer library, as implemented in Swiss-Prot PDB Viewer 4.1 software (Biozentrum, Basel, Switzerland) [39].

## 3. Results

Two unrelated Spanish cases with a clinical suspicion of KD, one early infantile and one adult, were referred to our laboratory for genetic analysis from the Pediatry and Neurology departments, respectively, of Hospital Universitario Ramón y Cajal.

### 3.1. Clinical Data

The first patient (42PET31 III:1) (Figure 1) was an eight-month-old male born to non-consanguineous parents. Neonatal development appeared normal until age 4 months, when his parents noticed regression in spontaneous movements, increasingly poor head support, and irritability. Clinical evaluation showed missed neurodevelopmental milestones, failure to thrive (below 2 SD in both height and weight), mild macrocephaly, laryngomalacia with dysphagia for liquids (which eventually required a gastrostomy, well tolerated, for proper feeding), mild hepatomegaly, and abnormally high levels of protein and lactate in the cerebrospinal fluid.

Magnetic resonance imaging (MRI), performed at age 7 months, revealed hypomyelination with unspecific lesions in the periventricular white matter (Figure 2). At age 8 months, neurophysiological analyses revealed extensive sensory and motor polyneuropathy. Whereas electroencephalography and visual evoked potentials (EP) were normal, brainstem auditory responses were asymmetric and suggestive of demyelination, and somatosensory EP were abnormal, with no recordable responses in the upper limbs and delayed and disorganized responses in the lower limbs.

The patient received symptomatic treatment (routinely: clonazepam, gabapentin, amitriptyline, selenium, paracetamol, vitamin D, and omeprazole; chlorpromazine and metamizole were added whenever intense restlessness/irritability appeared) and palliative care. He developed repeated apnea episodes that worsened whenever he had a respiratory infection and finally led to his death at age 3 years 9 months.

The second patient was a 33-year-old woman, PET199EXT II:1 (Figure 3), who presented with neurological symptoms. At age 17, she developed acute numbness and weakness in her left leg, with fluctuating intensity. Lumbar MRI and electromyography were unremarkable. Two months later, she presented with painless loss of vision in her right eye, with spontaneous recovery after 2–3 months. However, before complete recovery of vision in the right eye, she presented with similar symptoms in her left eye that eventually disappeared. A lumbar puncture displayed positive oligoclonal bands, with other autoimmune serologies being negative. Cranial MRI showed white matter T2 hyperintensities, misinterpreted as yuxtacortical and periventricular lesions. Multiple sclerosis was tentatively diagnosed, and treatment with interferon-beta was initiated.

After 10 years, she presented with clumsiness in her right leg when she underwent prolonged exercise. Suspecting a new relapse, her treatment was escalated to fingolimod. However, cranial MRI images presented an atypical evolution for multiple sclerosis, with marked hyperintensity of both pyramidal tracts, posterior periventricular white matter, and corpus callosum involvement without typical multiple sclerosis morphology (Figure 2). Alternative diagnoses were considered, and thus she was examined for accumulation of very-long-chain fatty acids in serum, for NMO-IgG and for MOG-IgG in serum (all tests being negative), and finally for KD, which was eventually confirmed by genetic and biochemical data. After atypical findings for multiple sclerosis, fingolimod was discontinued, without further relapses or lesions in the MRI. Neurological examination is unremarkable besides hyperreflexia, without other signs of pyramidal involvement.

Interestingly, the patient complained of eczematous lesions in her palms, which we identified as hyperlinearity (HPO 0033252) [40], i.e., exaggerated skin markings in the palm, typical of ichthyosis vulgaris and eczematous atopic dermatitis (Appendix A). Ichthyosis-like lesions have been reported in another individual with KD [41]; they are likely due to impaired ceramide synthesis in the skin caused by the galactosylceramidase enzymatic deficit that underlies KD. Indeed, treatment with a ceramide ointment led to improvement of the lesions within 1 month (Appendix A).

### 3.2. Biochemical Analyses

For patient 42PET31 III:1, biochemical analysis of DBS was performed at age 12 months by tandem mass spectrometry. The results indicated a deficit of galactosylceramidase activity (0.3 μmol/L/h; normal values between 0.4 and 4.1 μmol/L/h), typical of KD.

For patient PET199EXT II:1, tandem mass spectrometry assays on DBS samples also showed a clear deficit of galactosylceramidase activity (0.2 μmol/L/h; normal values between 0.7 and 7.3 μmol/L/h), consistent with a diagnosis of KD.

### 3.3. Genetic Analyses of Family 42PET31

We performed next-generation sequencing on the probands of both families with the resequencing panel STP v3, developed in our laboratory [18,19]. This panel targets 1821 regions from 101 genes and 174 different phenotypes from the Online Mendelian Inheritance in Man (OMIM) database [11], comprising 499 kb of genomic sequence.

Gene panel analysis of patient 42PET31 III:1 revealed a single heterozygous variant c.908+5G>A (depth: 185 reads; AF: 46%), at the beginning of intron 8 of the *GALC* gene (transcript NM_000153.4). We did not identify any other pathogenic variant in any of the genes targeted by the panel, including *PSAP*. We verified variant c.908+5G>A by Sanger sequencing in the proband and his parents; the variant was maternally inherited and shared by the maternal uncle of the patient (Figure 1). This variant has not been reported in any genomic databases so far. Following ACMG guidelines [25], we preliminarily classified it as a variant of uncertain significance (VUS; criteria PM2, PP3, and PP4) (Appendix A, [25]).

Analysis of variant c.908+5G>A with splicing prediction tool SpliceAI suggested loss of the splice donor site of intron 8 with high confidence (score 0.93). Thus, we synthesized cDNA by RT-PCR from RNA samples of carriers of the variant (II:2 and II:3) and also from II:1 to serve as a normal control. Amplification from exon 7 to exon 9 showed two bands in carriers of the variant and a single band in wild-type control individuals (Figure 4). The two bands were excised from a 2% agarose gel, purified, and sequenced. The smaller band corresponds to the skipping of exon 8 due to the predicted loss of the donor splice site of intron 8, while the larger band matches the expected wild-type transcript (Figure 4 and Figure 5). Thus, the skipping of exon 8 caused by the c.908+5G>A variant (c.753_908del) results in an in-frame deletion of 52 amino acids p.(Ala252_Ser303del). Taking into account the results of the RNA functional assays, we reclassified the c.908+5G>A variant as pathogenic as per ACMG guidelines (Table 1 and Appendix A).

As all clinical and biochemical data clearly indicated a galactosylceramidase deficiency, we sought the second putative variant (of paternal origin) underlying KD in patient 42PET31 III:1. We first searched for any deletions of the *GALC* gene in the heterozygous state that might have gone undetected by NGS heterozygosity and copy-number variation analyses. We performed an MLPA assay on genomic DNA, which ruled out that the patient harbored any deletions affecting *GALC*. Next, we hypothesized that the second variant could be a deep intronic change affecting splicing. As the death of the patient precluded obtaining any RNA samples, we sequenced the full-length cDNA obtained from his father’s RNA. Five overlapping PCRs were performed (Appendix A). Amplification of cDNA fragment 2 (exons 4 to 9) revealed a heterozygous insertion of a 34 bp sequence between exons 6 and 7 of the main *GALC* transcript, NM_000153.4 (Figure 5).

A review of known transcript variants in the NCBI database [42] showed two isoforms (NM_001424076.1 and NM_001424077.1) that include this cryptic exon. These two isoforms have a longer 5′-untranslated region comprising exons 1 to 6, as well as the cryptic exon included after exon 6. They encode a truncated polypeptide lacking the N-terminal 211 residues of canonical galactosylceramidase, and hence without catalytic activity, as it lacks the essential Glu-182 residue of the active center [32]. To determine whether the observed band with the insertion was just one of these minor isoforms or was indeed the product of an aberrant splicing event, we sequenced RNA from multiple unrelated controls. None of the controls showed the 34 bp insertion. Sequencing of the inserted 34 bp fragment in II:1 revealed a novel deep intronic variant c.621+772G>C, not found in any population databases. Analysis of the variant with SpliceAI predicted that both donor and acceptor splice sites for this alternative cryptic exon are strongly enhanced for recognition by the spliceosome (score changes: 0.07 to 0.51 for donor and 0.07 to 0.48 for acceptor). We designed primers to amplify the genomic region of intron 6 harboring this mutation (Appendix A), and segregation analysis confirmed that both the father and the affected son carried the novel variant. Inclusion of this cryptic exon (c.621_622ins621+758_621+791) in the main transcript leads to a frameshift and the appearance of a premature stop codon: p.(Ile208Valfs*31). Sequencing of the 593-bp wild-type transcript from the father showed that none of the exonic, harmless polymorphic variants associated in *cis* with the c.621+772G>C variant were present, which indicates that the c.621+772G>C variant leads to the inclusion of the 34-bp cryptic exon in nearly 100% of transcripts derived from this allele. Quantitative analysis indicated that the mutant isoform with the cryptic exon appears at much lower levels than the wild-type allele (Figure 5), most probably due to the triggering of nonsense-mediated decay in response to the appearance of the premature termination codon. Accordingly, we classified the novel variant c.621+772G>C as likely pathogenic (Table 1 and Appendix A).

### 3.4. Genetic Analyses of Family PET199EXT

STP v3 gene panel analysis of patient PET199EXT II:1 identified three heterozygous variants of interest in two different genes. At the *GALC* gene (transcript NM_000153.4), we detected one already known missense variant at exon 9 (depth: 342 reads, AF: 56%), c.956A>G (p.(Tyr313Cys)), and one novel splice-site variant at intron 9 (depth: 165 reads, AF: 47%), c.1034-1G>C. Segregation analysis by Sanger sequencing confirmed compound heterozygosity for both variants in the proband, with the father being a carrier of c.1034-1G>C and the mother of c.956A>G (Figure 3). The c.956A>G variant appears with a minor allele frequency (MAF) of 0.0000956 among genomes in the gnomAD database [23] and has been reported several times in KD patients of Asian descent [43,44,45], either homozygously or in the compound heterozygous state. Using cDNA synthesized from the proband’s RNA as a template, we performed PCR from exon 8 to exon 13 of *GALC*, identifying distinct bands corresponding to abnormal splicing events. When sequenced, the predominant smaller band corresponded to the skipping of exon 10 (c.1034_1161del) caused by the c.1034-1G>C variant (Figure 6), which results in a frameshift and the appearance of a premature termination codon, p.(Ala345Glufs*3). Considering all these data, assessment per ACMG guidelines classified c.956A>G as a VUS and c.1034-1G>C as pathogenic (Table 1 and Appendix A).

The remaining heterozygous variant detected by the gene panel analysis in the proband lay at the *GBA1* gene, underlying Gaucher disease, a different lysosomal disorder. It was the classic, known pathogenic Gaucher disease type 1 (visceral) mutation c.1226A>G, p.(Asn409Ser), previously termed N370S (depth: 382 reads, AF: 52%), which was inherited from her mother. To rule out any possibility that our patient could develop Gaucher disease in the future, we performed a rigorous clinical examination (but did not find any sign or symptom suggestive of Gaucher disease), and we also carried out further biochemical tests, as *GBA1* encodes the lysosomal hydrolase beta-glucocerebrosidase. The activity of this enzyme, assayed by tandem mass spectrometry, was within the normal range (4.2 μmol/L/h; normal values between 1.7 and 15.0 μmol/L/h), indeed ruling out Gaucher disease. Nevertheless, we did detect slightly higher than normal levels in plasma (9.4 ng/mL; normal values between 0.3 and 6.0 ng/mL) of glucosylsphingosine (Lyso-Gb1), a deacylated soluble metabolite accumulated at much higher levels by Gaucher disease patients [46]. Previous work showed that reduced levels of ceramide due to galactosylceramidase deficiency in KD patients causes reduced beta-glucocerebrosidase activity, through diminished maturation of the saposin C coactivator of beta-glucocerebrosidase [8]. Such maturation is catalyzed by cathepsin B and D lysosomal proteases, whose activity is dependent on ceramide levels [47]. Thus, the most likely explanation for the mild accumulation of Lyso-Gb1 observed in the plasma of PET199EXT II:1 is the interdependence of galactosylceramidase and those other lysosomal enzymes.

### 3.5. Structural Analysis of the Effects of GALC Variant p.(Tyr319Cys)

To better understand the pathogenicity of p.(Tyr319Cys), we intended to determine the effects of this amino acid substitution on the structure of galactosylceramidase. Other authors have reported X-ray diffraction structures of murine galactosylceramidase, either as a monomer [31], in complex with the galactosyl substrate [32], or as a heterotetramer made up of a central saposin A dimer with two monomers of galactosylceramidase bound to its surface [12]. However, to our knowledge, no structure of human galactosylceramidase (82.6% identical to the murine enzyme) has ever been published. For our analysis, we thus resorted to using a high-confidence (>94%) model of the human protein monomer, generated by AlphaFold2 and retrieved from the AlphaFold Protein Structure Database [36]. As a first step, we verified the accuracy of the model by performing pairwise structural comparisons between the model and all the available murine galactosylceramidase structures. In all cases, the Tyr-319 residue backbone and side chain of the model completely overlapped with those of the equivalent residue in the murine protein (Figure 7), indicating that we could safely proceed with the analysis of the structural effects of the p.(Tyr319Cys) amino acid substitution.

In the AlphaFold2 model, the –OH moiety of the aromatic ring of Tyr-319 forms a hydrogen bond with nitrogen in the guanidium side chain of residue Arg-396 within the same monomer. Replacement of the large aromatic ring of Tyr-319 with the smaller side chain of Cys would preclude the establishment of such a hydrogen bond because of the longer distance between the sulfhydryl group of Cys-319 and the guanidium of Arg-396 (Figure 7). Interestingly, Arg-396 is a non-catalytic residue involved in binding the galactosyl head group of the enzyme substrate within the active center of the hydrolase [32]. Indeed, Arg-396 alternates between two different conformations. In the first conformation, nitrogen in the guanidinium side chain of Arg-396 forms a hydrogen bond with the 6-hydroxyl of the galactose moiety, both at the beginning (substrate) and the end (product) of the catalytic cycle. The second conformation occurs in the unliganded enzyme and also upon formation of the covalent intermediate, when the side chain of Arg-396 moves back to the position observed in the unliganded enzyme [32]. Our analysis suggests that this second conformation in the human enzyme is stabilized by the hydrogen bond with the Tyr-319 residue.

## 4. Discussion

We report here three novel pathogenic variants that cause improper splicing of *GALC* transcripts: c.621+772G>C (inclusion of a cryptic exon between exons 6 and 7, resulting in p.(Ile208Valfs*31)), c.908+5G>A (skipping of exon 8, with an in-frame deletion of 52 amino acids), and c.1034-1G>C (skipping of exon 10, resulting in p.(Ala345Glufs*3)). The appearance of premature stop codons caused by the aberrant splicing events, along with the very low levels at which we observed the corresponding mutated transcripts, implies that c.621+772G>C and c.1034-1G>C would have a strong impact on *GALC* function due to the likely activation of nonsense-mediated decay. As for variant c.908+5G>A, its identification in a patient with early infantile KD also suggests a strong impact on galactosylceramidase function, even though the variant does not seem to activate nonsense-mediated decay.

The fourth variant identified in this study, c.956A>G, is a missense change replacing the Tyr-319 residue with cysteine. This allele is extremely rare in our population (MAF = 0.000443) [23], though it has been found in several South Asian populations at a maximum MAF of 0.017 (among Bengali from Bangladesh) [24]. Indeed, 14 Asian c.956A>G homozygotes, with undisclosed phenotypes, are reported among gnomAD v. 4.1.0 exomes [23]. Nevertheless, c.956A>G has been reported in 14 KD patients, mostly in the context of the milder, later-onset presentations, either in the homozygous state or in compound heterozygosity with other pathogenic variants (Appendix A, [43,44,45,48,49,50]). Visual impairment or walking difficulties were the initial signs and symptoms in all juvenile and adult cases, as observed in our patient.

The ClinVar database [20] reports “conflicting interpretations of pathogenicity” for variant c.956A>G, with three different laboratories classifying this variant as “benign” or “likely benign” and ten laboratories classifying it as “pathogenic” or “likely pathogenic”. In our case, strict application of ACMG guidelines classified c.956A>G as a VUS. However, previous functional analysis (expression from a transfected plasmid in COS-1 cells) did show a significant impact on galactosylceramidase activity (just 14% of the activity of the wild-type galactosylceramidase) [43]. Moreover, our molecular modeling suggests a role for the Tyr-319 residue in stabilizing the side chain of Arg-396, both in the unliganded enzyme and in one critical step of the catalytic cycle. Indeed, Arg-396 is essential for hydrolase function, as shown both by structural data [32] and by the fact that replacement of Arg-396 with either tryptophan [51] or leucine [52] results in severe KD of infantile onset. Thus, considering all available evidence, we conclude that the c.956A>G variant is a hypomorph that may cause KD, usually with milder phenotypes and later onsets (Appendix A). The strength of the variant in *trans* and the presence of other genetic modifiers [15] will usually determine the final KD presentation in individuals harboring c.956A>G, as just a small amount of residual galactosylceramidase activity will be enough to modify the phenotype dramatically [15].

One reason why c.956A>G may be deemed non-pathogenic is the fact that patients harboring it may sometimes test as false negatives in galactosylceramidase activity assays. Indeed, the activities of galactosylceramidase detected in leukocytes from the two c.956A>G compound heterozygotes from India reported by Gowda et al. [45] were normal. This phenomenon likely reflects the diverse binding affinities of the wild-type and the Cys-319 hydrolases for their natural substrates and for the artificial substrates used in the functional assays under cell-free experimental conditions. This is unsurprising given the close proximity of Tyr-319 to the active center residues that interact with the enzyme’s substrate, including Arg-396.

Activity assays on leukocytes or cultured fibroblasts are generally considered the gold standard for the definitive diagnosis of LDs caused by enzyme deficits. However, it must be kept in mind that these assays have their limitations, as shown in the false negatives discussed above and also in those false positives that are due to pseudodeficient alleles [53]. Consequently, KD diagnostic strategies should integrate all available clinical, biochemical, and genetic data, which is critical when facing late-onset KD cases with unusual presentations.

## Figures and Tables

**Figure 1 biomedicines-13-03114-f001:**
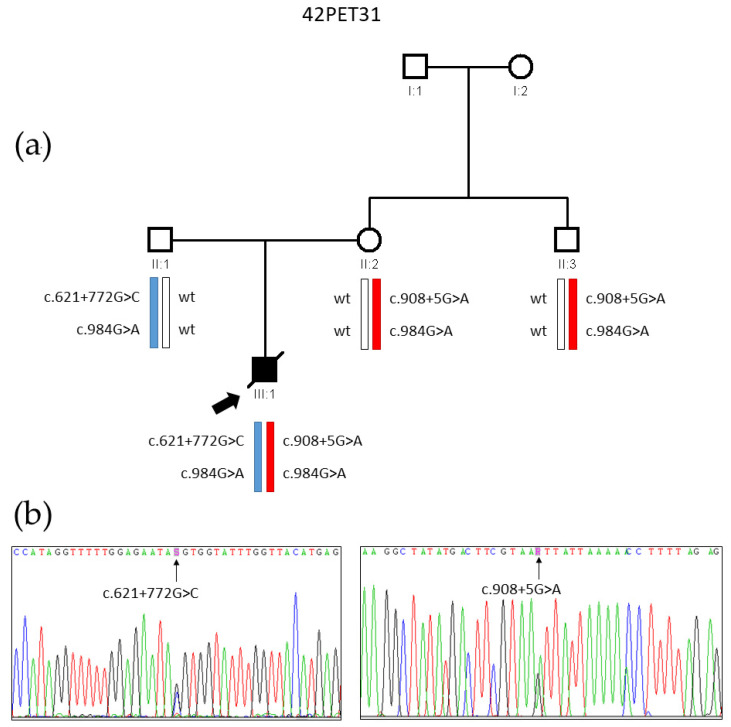
Genetic analysis of family 42PET31. (**a**) Pedigree of the family, showing the chromosomal segregation of *GALC* gene pathogenic variants and of the harmless polymorphic variant c.984G>A. Note that both pathogenic variants are in cis with the c.984G>A polymorphism, which was used to track the alleles in subsequent analyses. (**b**) Electropherogram showing the causal variants c.621+772G>C and c.908+5G>A, detected by Sanger sequencing of genomic DNA of proband III:1.

**Figure 2 biomedicines-13-03114-f002:**
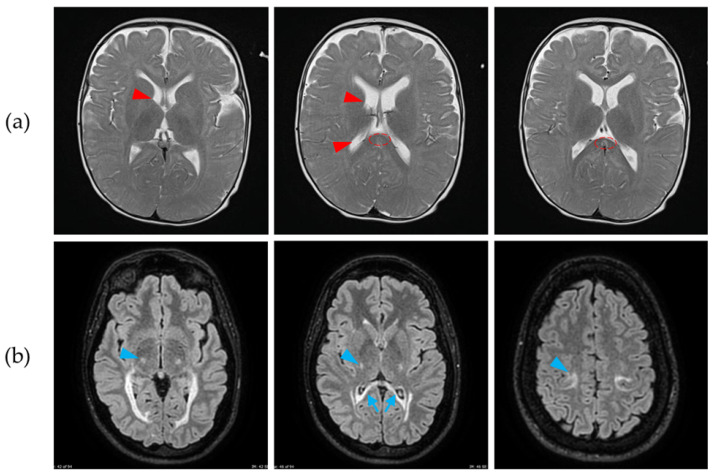
Magnetic resonance imaging (MRI) findings in the brain of our patients—axial sections. (**a**) In patient 42PET31 III:1, MRI at age 7 months showed an altered myelination pattern in the white matter of the internal capsules (arrowheads), with slight thinning of the splenium of the corpus callosum (dashed ellipse). We also observed hyperintensities in the periventricular white matter of both coronae radiatae. (**b**) In patient PET199EXT, we observed hyperintense T2-signal lesions involving the splenium of the corpus callosum (arrows), as well as the corticospinal tract (arrowheads). These findings have remained stable for more than 5 years.

**Figure 3 biomedicines-13-03114-f003:**
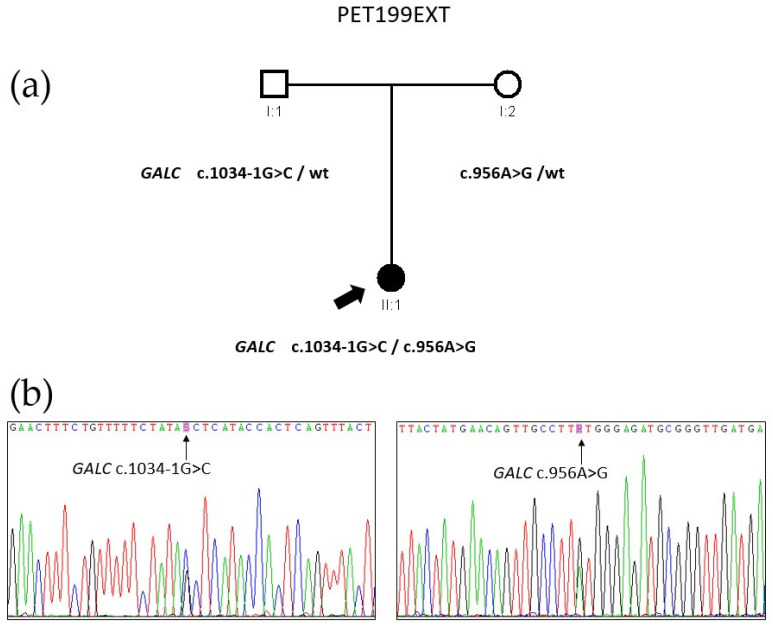
Genetic analysis of family PET199EXT. (**a**) Pedigree of the family, showing the chromosomal segregation of *GALC* pathogenic variants c.1034-1G>C and c.956A>G. Two additional variants were also identified in the *GBA1* gene: the known pathogenic variant c.1226A>G, and a synonymous variant c.708C>T with a low population frequency, for which RNA assays showed no pathogenic effect. (**b**) Electropherograms of the causal *GALC* variants in proband II:1 as observed by Sanger sequencing of her genomic DNA.

**Figure 4 biomedicines-13-03114-f004:**
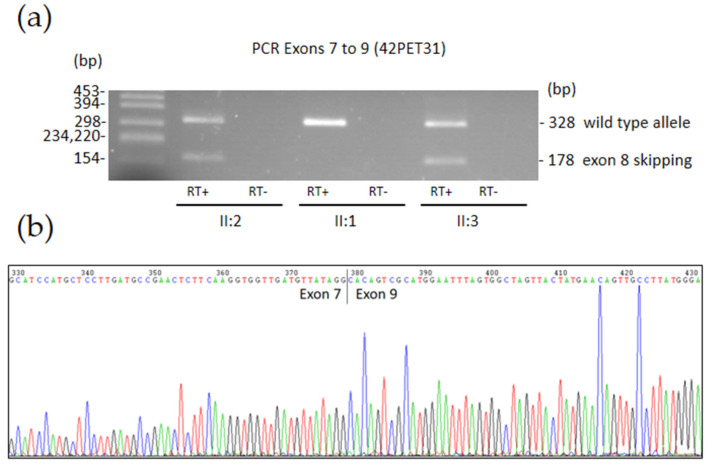
(**a**) Agarose gel electrophoresis of *GALC* cDNA amplification products spanning exons 7 to 9 from individuals II:2 (proband’s mother), II:1 (proband’s father), and II:3 (proband’s maternal uncle) of family 42PET31. A single band with the expected size (328 bp) for the correct splicing of exon 8 was obtained from II:1. In contrast, in carriers of the c.908+5G>A variant (II:2 and II:3), we observed both the 328-bp band (from the wild type allele) and an additional 178-bp band corresponding to aberrant splicing caused by skipping exon 8. RT+ indicates retrotranscribed RNA, and RT- refers to the RNA sample processed without reverse transcriptase, serving as a control to detect residual genomic DNA amplification. (**b**) Sanger sequencing electropherogram of the 178-bp band lacking exon 8.

**Figure 5 biomedicines-13-03114-f005:**
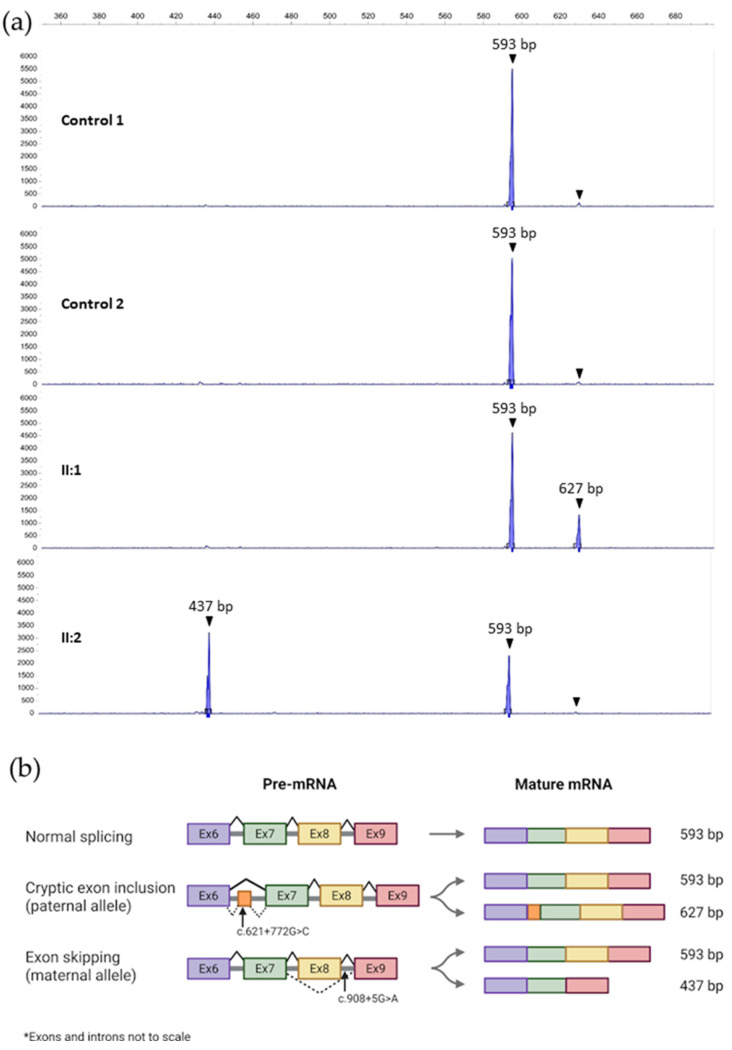
(**a**) Capillary electrophoresis of labeled cDNA amplification products spanning exon 6 to exon 9 of the *GALC* gene in two controls and the parents of patient 42PET31 III:1. The two aberrant splicing events detected are the inclusion of a cryptic exon in the father (caused by c.621+772G>C) and the skipping of exon 8 in the mother (caused by c.908+5G>A). The areas under the curve are equivalent to the amount of amplicon detected. Note a barely detectable peak with inclusion of the cryptic exon in control individuals without c.621+772G>C. (**b**) Representation of the aberrant splicing observed in family 42PET31 and transcript sizes. The first line corresponds to a wild-type individual with normal splicing. The second line shows the paternal alleles: one wild type and one with the deep intronic mutation c.621+772G>C. The third line shows the maternal alleles: one wild type and one with the splice site mutation c.908+5G>A. Panel (**b**) Created in BioRender by Domínguez Ruiz, M. 2025. Available online: https://BioRender.com/gkkcyvj (accessed on 10 October 2025).

**Figure 6 biomedicines-13-03114-f006:**
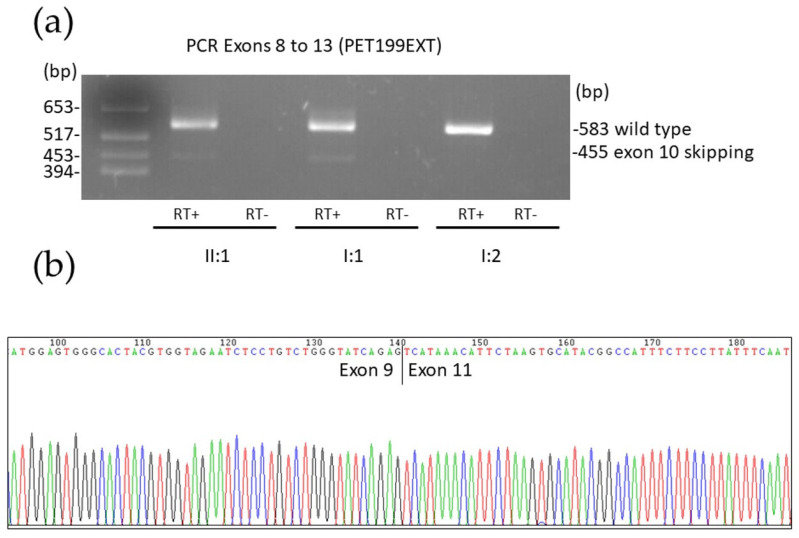
(**a**) Electrophoresis of *GALC* cDNA amplification products, spanning exons 8 to 13 in a 2% agarose gel, in the proband (II:1) and parents (I:1 father, I:2 mother) of family PET199EXT. A single band with the expected size (583 bp) for wild-type splicing was obtained from I:2. Carriers of the c.1034-1G>C variant in intron 9 (II:1 and I:1) present an additional 455 bp band corresponding to the skipping of exon 10. RT+ indicates retrotranscribed RNA, and RT- refers to the RNA sample processed without reverse transcriptase, serving as a control to detect residual genomic DNA amplification. (**b**) Sanger sequencing electropherogram of the 455-bp band lacking exon 10, with spliced exons 9 and 11.

**Figure 7 biomedicines-13-03114-f007:**
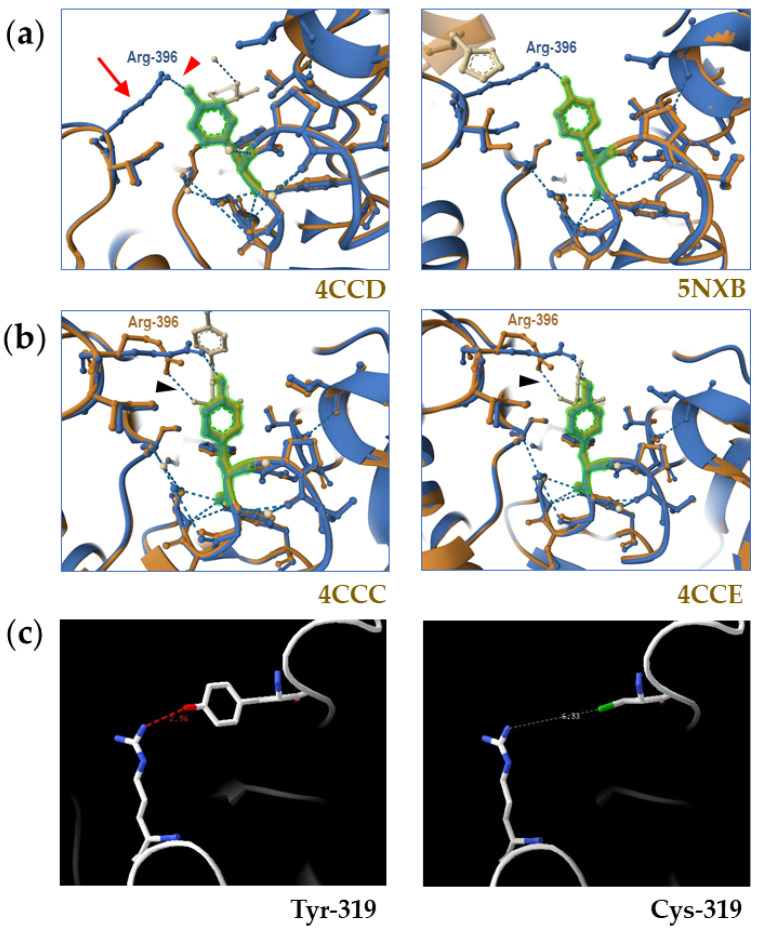
Analysis of the impact of *GALC* missense variant c.956A>G (p.(Tyr319Cys)) on the structure of galactosylceramidase. The AlphaFold2 model AF-P54803-F1-v4 of human galactosylceramidase always appears in blue in panels (**a**,**b**), while the different X-ray diffraction structures of murine galactosylceramidase appear in brown. (**a**) Pairwise comparison of the structure of the polypeptide chains of crystalized murine galactosylceramidase 4CCD and 5NXB and the AlphaFold2 model of human galactosylceramidase in the vicinity of residue Tyr-319 (highlighted in green), with a complete overlap of the side chain of Tyr-319 from the AlphaFold2 model with that of the X-ray diffraction structures. Please note the hydrogen bond (blue dashes and red arrowhead) between the –OH of the side chain of Tyr-319 and the nitrogen of the guanidinium side chain of Arg-396 (red arrow). In unliganded enzyme structures 3ZR5, 4CCD, and 5NXB, the side chain of the Arg-396 residue does not appear, indicating that it has different conformations in different copies of the crystal (i.e., it is highly mobile or disordered) and thus cannot be resolved clearly in electron density maps. (**b**) When the ligand (in ivory color) is bound to Arg-396 by a different hydrogen bond (black arrowhead), e.g., at the start and the end of the catalytic cycle of the hydrolase (structures 4CCC and 4CCE, respectively), the side chain of this residue (brown) is fixed, and it does appear in the structure (compare with panel (**a**)), without binding Tyr-319. (**c**) When Cys replaces Tyr at position 319, this hydrogen bond cannot form, because of the longer distance (2.96 versus 6.33 Å) between the sulfhydryl group of Cys-319 and the nitrogen atoms in the side chain of Arg-396.

**Table 1 biomedicines-13-03114-t001:** Assessment of pathogenicity of the variants identified in *GALC* (NM_000153.4).

Patient	Variant	CADD Score	Minor Allele Frequency	ACMG/AMP Criteria	Classification
42PET31 III:1	c.621+772G>C	27	0 (not found)	PS3, PM2, PM3, PP4	Likely pathogenic
42PET31 III:1	c.908+5G>A	20	0 (not found)	PS3, PM2, PM4, PP3, PP4	Pathogenic
PET199EXT II:1	c.956A>G p.(Tyr319Cys)	24	0.00044	PM1, PM3, PP3, PP5	Variant of uncertain significance
PET199EXT II:1	c.1034-1G>C	34	0 (not found)	PVS1, PS3, PM2	Pathogenic

## Data Availability

Data on the *GALC* pathogenic variants that we report in this study is available in the ClinVar database [20], with accession numbers SCV007096074 (c.621+772G>C), SCV007096076 (c.908+5G>A), SCV007103286 (c.956A>G) and SCV007096161 (c.1034-1G>C).

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
