# Peer review of "Abnormal Splicing of GALC Transcripts Underlies Unusual Cases of Krabbe Disease"

_biomedicines, 2025, doi:10.3390/biomedicines13123114_

Round 1
Reviewer 1 Report
Comments and Suggestions for Authors
The manuscript, "Abnormal Splicing of GALC Transcripts Underlies Unusual Cases of Krabbe Disease," submitted to the journal Biomedicines, analyzes two unrelated cases of Krabbe disease, with onset in early infancy and adulthood, respectively. The authors described the underlying pathogenic variants, demonstrated their pathogenicity, and discussed their impact on the clinical presentation. This work has significant implications for identifying early forms of Krabbe syndrome and developing effective treatments.
The authors meticulously analyzed the clinical data, meticulously conducted genetic studies, and performed molecular modeling, all of which are presented in the manuscript.
I have a few comments:
Lines 94-97: "KD is caused by biallelic pathogenic mutations in either the GALC gene (OMIM 606890) [11], encoding galactosylceramidase, or the PSAP gene (OMIM 176801), which encodes a precursor protein that is cleaved upon reaching the lysosome into four saposins, A through D."
Comment: It would be helpful to indicate the chromosomes on which these genes are located.
A question arose regarding the caption to Figure 2, which lists the patient's age as 7 months, whereas before the link to Figure 2 (lines 290-296), the author states the patient's age as 8 months.
Line 297: The patient received symptomatic treatment.
Comment: Please briefly describe the type of symptomatic treatment.
It would be helpful to expand on the abbreviation VUS in the caption to Figure 3. Lines 475-480: When comparing the structure of mouse and human galactosylceramidases, it is advisable to indicate the percentage of homology.
Author Response
Reviewer 1
The manuscript, "Abnormal Splicing of GALC Transcripts Underlies Unusual Cases of Krabbe Disease," submitted to the journal Biomedicines, analyzes two unrelated cases of Krabbe disease, with onset in early infancy and adulthood, respectively. The authors described the underlying pathogenic variants, demonstrated their pathogenicity, and discussed their impact on the clinical presentation. This work has significant implications for identifying early forms of Krabbe syndrome and developing effective treatments.
The authors meticulously analyzed the clinical data, meticulously conducted genetic studies, and performed molecular modeling, all of which are presented in the manuscript.
We thank Reviewer 1 for the kind endorsement on the quality of our manuscript.
I have a few comments:
Lines 94-97: "KD is caused by biallelic pathogenic mutations in either the GALC gene (OMIM 606890) [11], encoding galactosylceramidase, or the PSAP gene (OMIM 176801), which encodes a precursor protein that is cleaved upon reaching the lysosome into four saposins, A through D."
Comment: It would be helpful to indicate the chromosomes on which these genes are located.
We added the chromosomal locations on lines 97 and 98.
A question arose regarding the caption to Figure 2, which lists the patient's age as 7 months, whereas before the link to Figure 2 (lines 290-296), the author states the patient's age as 8 months.
MRI was performed at age 7 months, whereas the rest of the neurophysiological analyses were performed at age 8 months. Indeed, at age 8 months is when the clinical diagnosis of KD was established and the case was referred to our laboratory. This has been clarified now in the text (lines 292-298).
Line 297: The patient received symptomatic treatment.
Comment: Please briefly describe the type of symptomatic treatment.
This has now been added (lines 299-301)
It would be helpful to expand on the abbreviation VUS in the caption to Figure 3.
We have been unable to identify any reference to a VUS in the caption to Figure 3. However, we have expanded the VUS abbreviation used on Table 1 to its full “variant of uncertain significance”.
Lines 475-480: When comparing the structure of mouse and human galactosylceramidases, it is advisable to indicate the percentage of homology.
Human galactosylceramidase is 82.6% identical to the murine enzyme over its full length of 685 residues. This has been now added to the text (line 486).
We thank Reviewer 1 for all the helpful advice to improve our work.
Reviewer 2 Report
Comments and Suggestions for Authors
The manuscript "Abnormal splicing of GALC transcripts underlies unusual cases 2
of Krabbe disease" is en excellent incorporation of advanced modern methods into classical genetic research. Moreover, it has interesting medical implications. It is well written and beatifully illustrated. It is my pleasure to recommed accepting the manuscript for publication immediately.
Author Response
Reviewer 2
The manuscript "Abnormal splicing of GALC transcripts underlies unusual cases 2
of Krabbe disease" is en excellent incorporation of advanced modern methods into classical genetic research. Moreover, it has interesting medical implications. It is well written and beatifully illustrated. It is my pleasure to recommed accepting the manuscript for publication immediately.
Thank you very much for your appreciation of our manuscript as a valuable addition to the field.
Reviewer 3 Report
Comments and Suggestions for Authors
This research article reports finding of several novel pathogenic nucleotide variants contributing to pathogenesis of Krabbe disease (KD) via causing aberrant splicing of mRNA transcribed from GALC gene, which significantly affects activity of enzyme β-galactosylceramidase (GALC) encoded by this gene. The presence of aforementioned aberrantly spliced transcripts was confirmed by the authors. The low levels of “aberrant” transcripts in case of two novel nucleotide variants indicated nonsense-mediated decay. The authors also found pathogenic nucleotide variant in GBA1 gene – known to be involved in pathogenesis of Gaucher disease - in one case (in heterozygous state). Additionally, they performed in silico analysis (AlphaFold2 model) of impact of some of identified nucleotide variants on 3D structure of GALC. Further, authors gathered evidence that variant c.956A>G in GALC which they also detected in one of the cases is a hypomorph “that may cause KD, usually with milder phenotypes and later onsets”.
Please find below my comments and suggestions.
I recommend rewriting the abstract of the article, making it more “reader-friendly” for whose who are not specialists in KD. The recommended changes are minor. You may want to say something along the lines “Krabbe disease is a hereditary lysosomal disorder whose hall
mark is progressive demyelination with variable involvement of the central nervous system. It is caused by pathogenic nucleotide variants in the gene GALC encoding the enzyme β-galactosylceramidase (GALC), disrupting the function of this enzyme.”
Line 62 "pathogenic mutations", Line 69 "pathogenic <nucleotide> variants", Line 111 "harmful variants", Line 358 " harmful variant", etc - for the sake of consistency please use the same terminology throughout the text.
Line 124. Abbreviation for β-galactosylceramidase (GALC) is introduced only in Line 124. It should have been introduced earlier upon the first mention of β-galactosylceramidase in the text.
I also recommend introducing abbreviation KD (Krabbe Disease) in the very beginning of the text.
Name of the genes should be italicized.
Author Response
Reviewer 3
This research article reports finding of several novel pathogenic nucleotide variants contributing to pathogenesis of Krabbe disease (KD) via causing aberrant splicing of mRNA transcribed from GALC gene, which significantly affects activity of enzyme β-galactosylceramidase (GALC) encoded by this gene. The presence of aforementioned aberrantly spliced transcripts was confirmed by the authors. The low levels of “aberrant” transcripts in case of two novel nucleotide variants indicated nonsense-mediated decay. The authors also found pathogenic nucleotide variant in GBA1 gene – known to be involved in pathogenesis of Gaucher disease - in one case (in heterozygous state). Additionally, they performed in silico analysis (AlphaFold2 model) of impact of some of identified nucleotide variants on 3D structure of GALC. Further, authors gathered evidence that variant c.956A>G in GALC which they also detected in one of the cases is a hypomorph “that may cause KD, usually with milder phenotypes and later onsets”.
Please find below my comments and suggestions.
I recommend rewriting the abstract of the article, making it more “reader-friendly” for whose who are not specialists in KD. The recommended changes are minor. You may want to say something along the lines “Krabbe disease is a hereditary lysosomal disorder whose hall
mark is progressive demyelination with variable involvement of the central nervous system. It is caused by pathogenic nucleotide variants in the gene GALC encoding the enzyme β-galactosylceramidase (GALC), disrupting the function of this enzyme.”
This has been amended as suggested by the reviewer (lines 41-42).
Line 62 "pathogenic mutations", Line 69 "pathogenic <nucleotide> variants", Line 111 "harmful variants", Line 358 " harmful variant", etc - for the sake of consistency please use the same terminology throughout the text.
We have now used the terminology “pathogenic variant” throughout the text, as specified by the reviewer (e.g. lines 65, 71 and 362).
Line 124. Abbreviation for β-galactosylceramidase (GALC) is introduced only in Line 124. It should have been introduced earlier upon the first mention of β-galactosylceramidase in the text.
We have been careful not to use the GALC as an abbreviation for the enzyme galactosylceramidase to avoid confusion with the name of the gene (GALC), even though the gene name is italicized. Thus, we have removed from the text this single appearance of the GALC abbreviation for the enzyme name (line 127).
I also recommend introducing abbreviation KD (Krabbe Disease) in the very beginning of the text.
We have introduced the KD abbreviation in the abstract as suggested (line 39).
Name of the genes should be italicized.
This has been revised and corrected throughout the text.
We thank Reviewer #3 for the insightful comments and helpful suggestions.